# Impact of phage enrichment on the observed infant gut phageome

Sandro Valenzuela-Diaz,[1] Evgenia Dikareva,[2] Brandon Hickman,[1] Saija Kiljunen,[1] Kaija-Leena Kolho,[1,3] Willem de Vos,[1] Anne Salonen,[1] Katri Korpela[1,4]

**ABSTRACT** The human gut microbiota, particularly during infancy, plays a pivotal role in shaping long-term health outcomes. While research on the bacterial microbiota has advanced rapidly, the infant gut virome—dominated by bacteriophages—remains underexplored due to technical challenges in viral DNA detection and recovery. To address this, we optimized a polyethylene glycol (PEG)-based protocol for phage DNA enrichment tailored to the constraints of infant fecal samples, focusing on maximizing viral yield from minimal input material. We validated the optimized protocol on fecal samples from 41 infants at 1, 6, and 12 months of age and assessed the impact of phage enrichment on the observed gut phageome. The results demonstrate that the optimized protocol improves viral DNA recovery and significantly alters the observed virome composition, especially in older infants. Without appropriate enrichment, key features of the gut virome may be underrepresented or missed entirely. These findings underscore the importance of protocol optimization in virome studies and provide a scalable, cost-effective method for robust infant gut virome profiling.

**IMPORTANCE** Understanding the viral component of the infant gut microbiome is essential for uncovering its role in early-life health, yet technical limitations have hindered its study. This work presents a systematically optimized and validated protocol for enriching viral DNA from infant stool samples, designed specifically for low-input material typical of early life. By adapting polyethylene glycol-based precipitation methods, we achieved consistent and scalable recovery of viral DNA across infants of different ages. Application of this protocol revealed key age- and delivery mode-specific differences in phage diversity and replication strategies that were undetectable using standard approaches. Our findings demonstrate that careful protocol optimization is critical for accurate virome profiling in infants and offer a practical solution to overcome longstanding methodological challenges in the field.

**KEYWORDS** gut microbiota, virome, viral particle enrichment, PEG, DTT, LMM

The human gut microbiota, a dynamic community of microorganisms, has attracted significant attention for its profound impact on human health, particularly during early life stages (1) when the microbiota is established. Numerous studies have highlighted the importance of the infant gut microbiota in shaping long-term health outcomes (2–4). Even minor perturbations during this formative phase can lead to lasting health consequences (1, 5).

Despite advancements in our understanding of the bacterial microbiota, much less is known about the gut virome, which encompasses eukaryotic viruses and bacteriophages (6). Bacteriophages, or phages, represent the most abundant entities within the human gut microbiota, collectively referred to as the gut phageome (7). Gut phages have gained significant attention for their crucial role in shaping microbial dynamics, enhancing

Address correspondence to Katri Korpela, katri.korpela@helsinki.fi.

The authors declare no conflict of interest.

genetic diversity, and maintaining ecosystem stability within the gut (8, 9). Recent efforts, including the development of specialized databases and compendiums, have expanded our understanding of these phages (10–13).

Although general knowledge of phage ecology has grown, detailed insights into phage-bacterium and phage-host interactions remain limited (14). Detecting phages is essential to understanding their ecological role, and next-generation sequencing (NGS) has become a key tool for this purpose (15). However, challenges persist, including low viral DNA yields, difficulty in phage DNA extraction, and high viral diversity, which limits taxonomic and functional annotation (7, 11, 13, 16). Addressing these issues in the laboratory requires more advanced tools for phage identification and quantification.

Various protocols have been developed to enrich phage DNA from fecal samples, with dithiothreitol (DTT) (17) and polyethylene glycol (PEG) precipitation (7) being the most used. While effective with NGS, these methods require adaptation for large-scale studies, especially in infants where sample volume is limited. To address this, we present an optimized, high-throughput phage DNA enrichment protocol and compare virome profiles from enriched and non-enriched infant fecal samples to assess its impact on the observed phageome.

## MATERIALS AND METHODS

### Sample collection

Samples were collected from 41 infants belonging to the HELMI-Plussa study, where the participants are part of the Finnish Health and Early Life Microbiota (HELMi) longitudinal birth cohort (18). Samples were collected daily for a period of 20–30 days during the infant's 6th and 12th months. Some infants were sampled at both time points, but the different time series were treated as separate. In total, we had 1,271 daily fecal samples, 506 samples at 5 to 6 months and 765 samples at 11 to 12 months. Samples were stored at −20°C in the home freezer, transported on dry ice, and stored at −80°C upon arrival to the laboratory.

### Phage enrichment and DNA extraction protocols

We first tested five protocols for viral enrichment (named D1, D2, P1, P2, and P3) on a subset of 54 infant fecal samples based on two commonly used extraction methods: DTT (17) and PEG precipitation (7). These methods are widely used in virome studies and considered standard approaches for phage enrichment (PE) from fecal material due to their effectiveness in lysing host cells (DTT) and concentrating viral particles (PEG). A schematic overview of the five protocols is shown in Fig. 1A, highlighting the shared workflow and the specific modifications tested between DTT- and PEG-based approaches.

To adapt the PE protocol for infant fecal samples, we performed a series of optimization steps focused on achieving reliable DNA recovery from limited starting material. Given the small volume of fecal matter available from infants, we scaled down the original protocol requirements while preserving its functional integrity, that is, retaining all critical steps of the protocol—lysis of host cells, removal of host DNA, concentration of viral particles, and nuclease protection—in their original sequence. Only the sample input and reagent volumes were reduced proportionally, and incubation times and enzymatic treatments were kept unchanged. Additionally, three different buffers were tested to measure the impact on the final DNA concentration. The buffer compositions used were as follows:

- SMG1 buffer (200 mM NaCl, 10 mM MgSO4, 50 mM Tris-HCl [pH 7.5] and 0.01% gelatin)

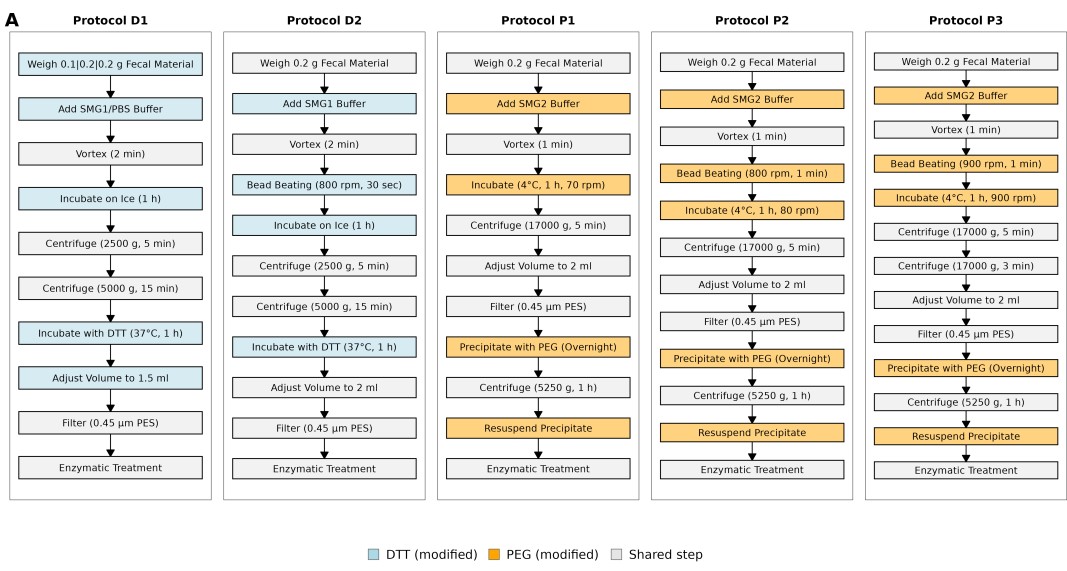

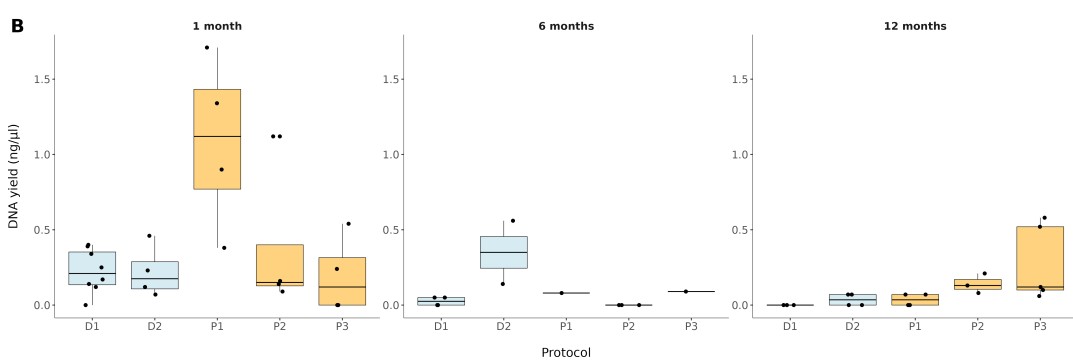

FIG 1 (A) Overview of the five viral DNA extraction protocols (D1–D2, PEG1–PEG3). The flowchart illustrates the main workflow steps shared across protocols (weighing of fecal material, buffer addition, homogenization, incubation, centrifugation, filtration, DNase I treatment, and DNA isolation) as well as the specific modifications tested, including different buffers, beating steps, incubation conditions, and centrifugation settings. Protocols D1–D2 are based on DTT treatment, while protocols P1–P3 rely on PEG precipitation. (B) DNA yield (ng/µL) comparison across five different PE protocols at three time points: 1 month, 6 months, and 12 months. Y-axis represents the DNA yield (ng/µL), and the x-axis shows the protocols as their color.

- SMG2 buffer (100 mM NaCl, 10 mM MgSO4, 50 mM Tris-HCl [pH 7.5], 0.01% gelatin)
- PBS buffer (phosphate buffered saline, pH 7.00).

## DTT DNA extraction method (protocols D1 and D2)

First, we aimed to standardize the amount of wet fecal material; thus, for protocol D1, we used amounts of 0.1, 0.2, and 0.5 g. The starting material of 0.1 g of feces yielded 0–0.25 ng of DNA, 0.2 g of fecal material yielded 0–0.39 ng of DNA, and 0.5 g of fecal material yielded 0–0.05 ng of DNA. As increasing the amount of fecal matter did not increase the DNA yield, further protocols were continued with 0.2 g (Fig. S1).

For protocol D1, fecal material was added in 1.5 mL of SMG1 buffer, then vortexed for 2 min, incubated on ice for 1 h, and centrifuged twice: first for 5 min, at 4°C, 2,500 × $g$ and then for 15 min, at 4°C, 5,000× $g$. The supernatant was incubated with 6.5 mM DTT (Thermo Fisher Scientific, Life Technologies Europe, Netherlands) at 37°C for 1 h. Then, the sample volume was adjusted to 1.5 mL with SMG1 buffer. The mixture was filtered with 0.45 µm polyether sulfone (PES) filter (VWR, USA), to which DNase I (DNase I kit, RNase-free, Thermo Scientific, Lithuania) was added to a final concentration of 0.01 U/µL. The samples were incubated for 1 h at 37°C. The reaction was stopped by addition of ethylenediaminetetraacetic acid (EDTA, provided with DNase I kit, Thermo Scientific,

Lithuania) to a final concentration of 5 mM and heated for 10 min at 65°C. During the DNA extraction, the samples were treated with Proteinase K (Thermo Scientific, Lithuania) to a final concentration of 0.05 mg/mL for 30 min at 55°C. The phage DNA was extracted with the Norgen Phage DNA isolation kit (Norgen Biotek Corp., Canada) according to the manufacturer's instructions. The concentration of eluted DNA was determined with Qubit Fluorometer and Qubit dsDNA HS Assay Kit (Thermo Fisher Scientific, USA) and stored at −80°C.

Protocol D2 was the same as protocol D1 with the following changes: 0.2 g of fecal material was mixed with 1.5 mL of SMG1 buffer, a beating step (800 rpm, 30 s, on FastPrep-96 [MP Biomedicals] instrument) was added after the vortex step, and the sample volume was adjusted to 2 mL before the filtering step. During the DNA extraction with the Norgen Phage DNA isolation kit, the elution volume was reduced from 70 to 45 µL. These modifications were introduced to improve homogenization of the fecal matrix and to increase the DNA concentration in the final eluate.

### PEG DNA extraction method (Protocols P1, P2, and P3)

In protocol P1, 0.2 g of wet fecal material was added to 1.5 mL of SMG2 buffer. After 1 min of vortexing, the samples were incubated at 4°C for 1 h, 70 rpm and then centrifuged for 5 min at 4°C, 17,000× $g$. The volume of the collected supernatant was adjusted to 2 mL with SMG2 buffer and filtered through 0.45 µm PES filter. Then, the samples were precipitated overnight at 4°C with 10% PEG (PEG8000, Fluka, BioChemika) and 1 M NaCl (Fisher Chemical). After the incubation, the samples were centrifuged for 1 h at 4°C, 5,250× $g$, and the precipitate was resuspended in 325 µL of SMG2 buffer. The DNase treatment was the same as in protocol 1, except the incubation time was only 30 min instead of an hour. The phage DNA was extracted with the Norgen Phage DNA isolation kit (Norgen Biotek Corp., Canada) according to the manufacturer's instructions.

Protocol P2 was the same as protocol P1 with the following changes: a faster homogenization step (a beating step, 800 rpm, 1 min) was added after the vortexing step and the speed was raised from 70 to 80 rpm during the 1-h incubation step at 4°C. The aim of these changes was to enhance viral particle release from the fecal material and to improve precipitation efficiency during PEG treatment.

Protocol P3 was adapted from protocol P2 as follows: Commercial tubes with beads of size between 70 and 125 microns in diameter (Bead tubes, Purelink, Invitrogen) were used in the homogenization step to perform bead beating instead of normal Eppendorf tubes, the speed was raised from 80 rpm to 900 rpm during the 1-h incubation step at 4°C and centrifugation time was increased from 5 to 8 min. These adjustments were made to maximize disruption of the fecal matrix and to increase the recovery of viral particles in samples with more complex composition.

### Whole-metagenome DNA extraction

For whole metagenome extraction, DNA was extracted from stool using a bead-beating method. Approximately 250 or 340 mg of wet fecal material was suspended in 0.5 or 1 mL of sterile ice-cold PBS, and 250 µL of the fecal suspension was combined with 340 µL of RBB lysis buffer (500 mM NaCl, 50 mM Tris–HCl [pH 8.0], 50 mM EDTA, 4% SDS) in a bead-beating tube from the Ambion MagMAX Total Nucleic Acid Isolation Kit (Life Technologies). Bead-beating was performed in two consecutive cycles, following the manufacturer's recommendations, to ensure efficient cell disruption. After bead-beating, 200 µL of the supernatant was used for DNA extraction with a KingFisher Flex automated purification system (ThermoFisher Scientific) using the MagMAX Pathogen RNA/DNA (high volume) program. DNA concentrations were determined using the Quant-iT PicoGreen dsDNA Assay (Invitrogen).

## Sequencing

Samples processed exclusively with standard metagenomic extraction were designated as the baseline group (Control), whereas samples subjected to both metagenomic extraction and viral particle enrichment were classified as phage-enriched (PE). The DNA from the viral particles and the whole metagenome was pooled at equal concentrations for library preparation to be able to sequence both bacterial and viral DNA fractions together. Libraries were prepared using the Illumina DNA Prep kit (19) and sequenced using Illumina NovaSeq (150 bp ×2 at an average of 25 ± 8 Gb) at the Institute for Molecular Medicine Finland (FIMM).

## Bioinformatic approach

### Preprocessing

Raw reads were merged and filtered using fastp v0.24.0 (20). Reads that could not be merged were treated as single reads. To remove host DNA, filtered reads were mapped using Minimap2 v2.24 (21) against the human genome (GRCh38.p14, NCBI RefSeq assembly: GCF_000001405.40) and SAMtools v1.21 (22) to recover the non-mapped reads. On average, 84.94% ± 5.14% (mean ± sd) of total reads passed these filters. Details of parameters can be found in Fig. S2 in the supplementary material.

Filtered reads were gathered by time series ID (infant ID and age) and assembled using MEGAHIT v1.29 (23). Contigs shorter than 2,000 bp were discarded to meet minimum software input requirements using custom Perl script (removesmalls.pl in Supplementary material).

### Mining

The phage mining started by mapping the contigs using blast v2.16 (24) against the Unified Human Gut Virome Catalog (UHGV) (10, 12, 13, 21, 22, 25–30) available at https:// github.com/snayfach/UHGV. This database was also used to create an unphaged version of the HumGut database (HGDB) (23) in order to avoid future false positives while removing bacterial contigs from the assemblies. The unphasing process was carried out using blast v2.16 (24) at 70% of coverage and identity. Then, viral coordinates detected in the bacterial contigs were removed, leading to a split bacterial contig (seqcleaner.py in Supplementary material). In total, 230,098 prophages were removed from the HumGut database (HGDB).

To extract prophages from bacterial contigs in our metagenomics data, assemblies were subjected to CheckV v1.03 (31), PhageBoost v1.0.3 (25). The contigs used for prophage extraction were also mapped against the unphaged version of the HGDB using Kraken2 v2.1.5 (32) to annotate the host bacteria. Contigs identified as bacteria were filtered out using a custom R script v4.3. Parameters used can be found in Fig. S2 in the supplementary material and in slurm_scripts/3_prophage.sh.

The remaining contigs were analyzed for binary phage classification using Meta-PhaPred v1 (33) under default parameters and PhaMer v2.1.10 (28), with a score threshold of 0.9 in a post-parsing process. Subsequently, the remaining contigs underwent binary viral classification using DeepVirFinder v1.0 (34) and VirSorter2 v2.24 (35). Then, to remove false positive phage predictions, a minimum score agreement greater than one was applied by adding Phamer and MetaPhaPred scores. Finally, phage contigs were dereplicated at 98% identity and 85% of coverage.

### Annotation

Host prediction on viral contigs was performed by iPhop v1.3.3 (36). A post-processing was performed to keep only high-quality predictions with a cutoff score of 95 and filtering out predictions when no shared proteins were reported (r_scripts/ parse_hostpred.R in Supplementary material). The resulting predictions were added to the annotations performed by Kraken2 v2.1.5 in the prophage extraction step (Fig. S2).

Contigs were binned using vRhyme v1.1.0 (37) under default parameters. A second round of binning was performed for contigs with no bin assigned using ALFATClust v1 (38).

Taxonomy assignment was done with PhaGCN v2.1.10 (39). A postprocess using a rejection score of 0.7 at family level was performed (r_scripts/parse_phagcn.R in Supplementary material). Additionally, to avoid false environmental classification, we filtered out taxa not found in the UHGV taxonomy. ViroTaxo v2 (40) (default parameters) was also used for taxonomy predictions. The software was previously trained with the UHGV database using the build script the developers provided. Taxonomy assignments were performed individually by contigs and then on the bin-level by propagating the taxonomy to the rest of the members in the bin. Bins with different taxonomies were discarded from further analyzes, representing only 1% of the total annotation. Finally, phage lifestyle was predicted using PhaTYP v2.1.10 (39) using a rejection score of 0.7.

## Calculation

Raw read abundance was calculated by mapping each sample to its corresponding virus-mined contigs using Minimap2 v2.24 and SAMtools v1.21, discarding non-mapped reads, supplementary and secondary alignments (slurm_scripts/getvircovs.sh in Supplementary material). Additionally, to address the under-representation of viral abundances in metagenomic samples, we further calculated viral relative units using the following formula:

$$\mathrm{VRU}_i = \frac{\mathrm{mdepth}_i \cdot \mathrm{coverage}_i}{\sum_{i=1}^{n} \mathrm{mdepth}_i \cdot \mathrm{coverage}_i}$$

*VRU* = viral relative units,
*i*: is the *i*-th viral contig,
mdepth: mean depth on mapped based, and
coverage: percentage [0,1] of the contig covered.

Under this approach, a total of 33,055 contigs were mined, from which annotations reached 34% for taxonomy at family level, 77% for lifestyle, and 78% for host prediction at genus level. Phage contigs were finally summarized into groups by summing together contigs with the same taxonomy, lifestyle, and host.

## Statistical analysis

To assess phageome diversity, we calculated the alpha diversity as the Shannon diversity index using the alpha.div() function in R package asbio v1.9-7 on predicted family phages.

To identify key sources of variation and reduce the dimensionality of the phageome data, we performed a principal coordinates analysis (PCoA) on the viral family-level relative abundance data. This analysis was conducted using the capscale() function from the vegan package in R (version 2.6-8) and Pearson correlation as the distance metric. Before conducting the PCoA, we normalized the viral relative units' data through log-transformation to address large differences in relative abundance among viruses. Finally, to evaluate the significance of various infant factors, such as individual ID, age, and birth mode, we used the adonis() function from the vegan package.

To assess the impact of PE on the observed phages (presence/absence) by taking into consideration the individuality of the infant, we fitted a linear mixed-effect model (LMM) through the lmer() function from lme4 R package (41) to test the impact of the phage enriched protocol under the following formula:

$$\log(Y_{ij}) = \beta_0 + \beta_{\mathrm{treatment}} * \mathrm{treatment} + R_j + \varepsilon_{ij}$$
$$P(Y_{ij} = 1) = \beta_0 + \beta_{\mathrm{treatment}} * \mathrm{treatment} + \beta_{\mathrm{delivery}} * \mathrm{delivery} + \beta_{\mathrm{age}} * \mathrm{age} + R_j + \varepsilon_{ij}$$

$P(Y_{ij} = 1)$: probability of the presence of the ͡i virus/host in the j group,

$\beta_0$: intercept (average log-transformed abundance when all other predictors are zero),

$\beta_{treatment}$: effect of the PE protocol in viral abundance,

$\beta_{delivery}$: effect of the delivery in viral abundance,

$\beta_{age}$: effect of the age in viral abundance,

$R_j$: random intercept for the $j$ group (baby ID), and

$\varepsilon_{ij}$: residual term for the $j$ virus in the $j$ group.

The latter model was separately fitted to the time series with and without the PE to assess the impact of PE on the observed effect of birth mode and age on the phageome.

## RESULTS

In this study, we collected fecal samples from seven cesarian born and 34 vaginally born infants, resulting in 216 and 1,055 samples, respectively. We grouped the infant fecal samples into two time windows based on the infants' chronological ages at the time of collection. A median of 25 ± 6 (mean ± sd) daily fecal samples were collected from each infant. In total, 1,271 fecal DNA samples from 41 infants were subjected to deep metagenomic sequencing (150 bp paired-end reads). Of these samples, 340 samples were additionally subjected to a PE protocol before pooling the viral DNA with the metagenomic DNA and subsequent co-sequencing of the phageome and bacteriome. The sample details can be found in Table S2.

### DNA yield

To develop an effective PE protocol suitable for infant fecal samples, we first conducted a pilot optimization study on a subset of samples ($N = 54$), including those from infants as young as one month. The amount of starting material was limited from the outset, making DNA recovery a consistent challenge. Our primary objective was therefore to refine a scalable protocol capable of maximizing viral DNA yield despite these constraints across different infant age groups. Further details can be found in Table S1.

Throughout the pilot optimization process, we observed substantial variation in DNA yield depending on the procedural adjustments (Fig. 1B). At 1 month of age, protocol P1 produced the highest yields, reaching up to ~1.6 ng/µL, followed by protocol P3 with ~0.5 ng/µL. At 6 months, DNA recovery was markedly lower for most of the protocols (<0.2 ng/µL). By 12 months, the optimized PEG-based protocol P3 outperformed the others, yielding up to ~0.5 ng/µL, while all remaining protocols stayed close to background levels (≤0.1–0.2 ng/µL). When comparing the ratio of mean yield to its variability (mean/SD), protocol P3 achieved the highest stability (0.97) relative to the other protocols (D1 = 0.85, D2 = 0.90, P1 = 0.78, P2 = 0.58). Although P1 occasionally produced the highest absolute yields in early samples, P3 was the only protocol that maintained reproducible performance across ages and was therefore selected for subsequent analyzes. Finally, no clear taxonomic differences between protocols were observed in the pilot, and the selection of P3 for subsequent analyses was therefore based on yield and reproducibility across ages. However, we did not systematically evaluate potential taxonomic biases, which remain an inherent limitation of any enrichment method.

### Impact of PE on the infant gut phageome

To validate the optimized PE protocol (P3), we assessed its influence on the infant gut phageome in relation to age and delivery mode. Samples processed with both the optimized PE protocol and standard metagenome extraction (capturing virome and bacteriome) were compared to baseline samples subjected only to standard metagenome extraction. Phage reads represented 6.09% ± 2.78% of total reads in control samples and 6.57% ± 4.86% in PE-treated samples. LMM, with treatment as the fixed effect, did not reveal an overall difference in viral read relative abundance ($P > 0.05$, Table S3). However, by stratifying by born method, a significant increase in viral reads was observed in cesarean-born infants at both 6 and 12 months in PE-treated samples

($P$ < 0.05 and $P$ < 0.01, respectively; Fig. 2A), indicating that phage detection in certain subgroups may be particularly sensitive to enrichment.

Across all samples, the infant gut phageome observed comprehended 36 viral families, averaging 11 per infant. PE samples at 6 months showed increased family diversity compared with control samples at the same age ($P$ < 0.05, Fig. 2C), even though the number of viral contigs was similar (Fig. 2B). In control samples, viral diversity increased with age (Fig. 2C), but this trend was not significant in PE samples ($P$ > 0.05, Table S3). Ordination showed clustering by infant (Fig. S3), with infant ID and age explaining 89%–97% of variance ($P$ = 0.001, Adonis test with Pearson correlation distance) and PE accounting for 3–11% of variance (Fig. S3).

## Impact of PE on the viral lifecycle

To explore the replication dynamics of the infant gut phageome, we classified 25,327 contigs (76.62%) as either virulent or temperate. Virulent phages comprised a substantial portion of the total phage abundance (32% ± 13%, mean ± SD). LMM analysis revealed that temperate phages were consistently more abundant than virulent ones, irrespective of infant age or delivery mode ($P$ < 0.001, Table S3).

We then assessed the extent to which the optimized PE protocol influenced the detection of different phage replication strategies. Among vaginally delivered infants, no significant differences were found between control and PE-treated samples in the relative abundance of virulent or temperate phages at either 6 or 12 months ($P$ > 0.05, Fig. 3). However, in cesarean-born infants, PE treatment revealed a significantly higher relative abundance of virulent phages at both time points ($P$ < 0.001, Fig. 3). These findings suggest that standard metagenomic approaches may underestimate the presence of free virulent phages in certain subpopulations, particularly cesarean-born infants.

Further LMM analysis indicated age-related changes in virulent phage abundance within the control group of cesarean-born infants, with a notable decline from 6 to 12 months ($P$ < 0.001, Table S3). No such trend was observed among vaginally delivered infants ($P$ > 0.05). At 12 months, control samples from vaginally born infants exhibited significantly higher virulent phage abundance than those from cesarean-born infants ($P$ < 0.001), a difference that was not evident at 6 months.

### PE increases phage diversity.

We classified 11,113 contigs (34%) to the family level, which were then used to assess virome diversity. As expected, taxonomic diversity increased with infant age (Fig. S4). In vaginally delivered infants, the application of the optimized PE protocol resulted in significantly higher alpha diversity at 6 months compared to controls ($P$ < 0.05, Fig. 4A). At 12 months, this effect was not observed ($P$ > 0.05, Table S3). Among cesarean-born infants, PE treatment yielded a significant increase in alpha diversity at 12 months ($P$ < 0.05, Fig. 4A), bringing diversity levels closer to those of vaginally born counterparts. Notably, in control samples from vaginally born infants, diversity was positively associated with age ($P$ < 0.05, Table S3), while this association was not significant in PE-treated samples ($P$ > 0.05, Table S3).

To further evaluate the influence of enrichment on phage detection, we applied a binomial LMM across 36 viral families stratified by predicted life cycle. We defined family presence as any relative viral unit greater than zero. Although the PE treatment had a limited direct effect on the presence of individual families, delivery mode and infant age showed stronger influences on phage family prevalence (Fig. 4B; Table S3). These findings suggest that while the PE protocol enhances overall diversity captured, the detection of specific phage taxa is primarily shaped by biological variables.

To assess whether the optimized PE protocol influences the detection of age- and birth mode-associated differences in phage prevalence, we applied LMM separately to control and PE-treated groups. While the initial model included all variables simultaneously, stratified analyses revealed that PE treatment markedly altered the detection of

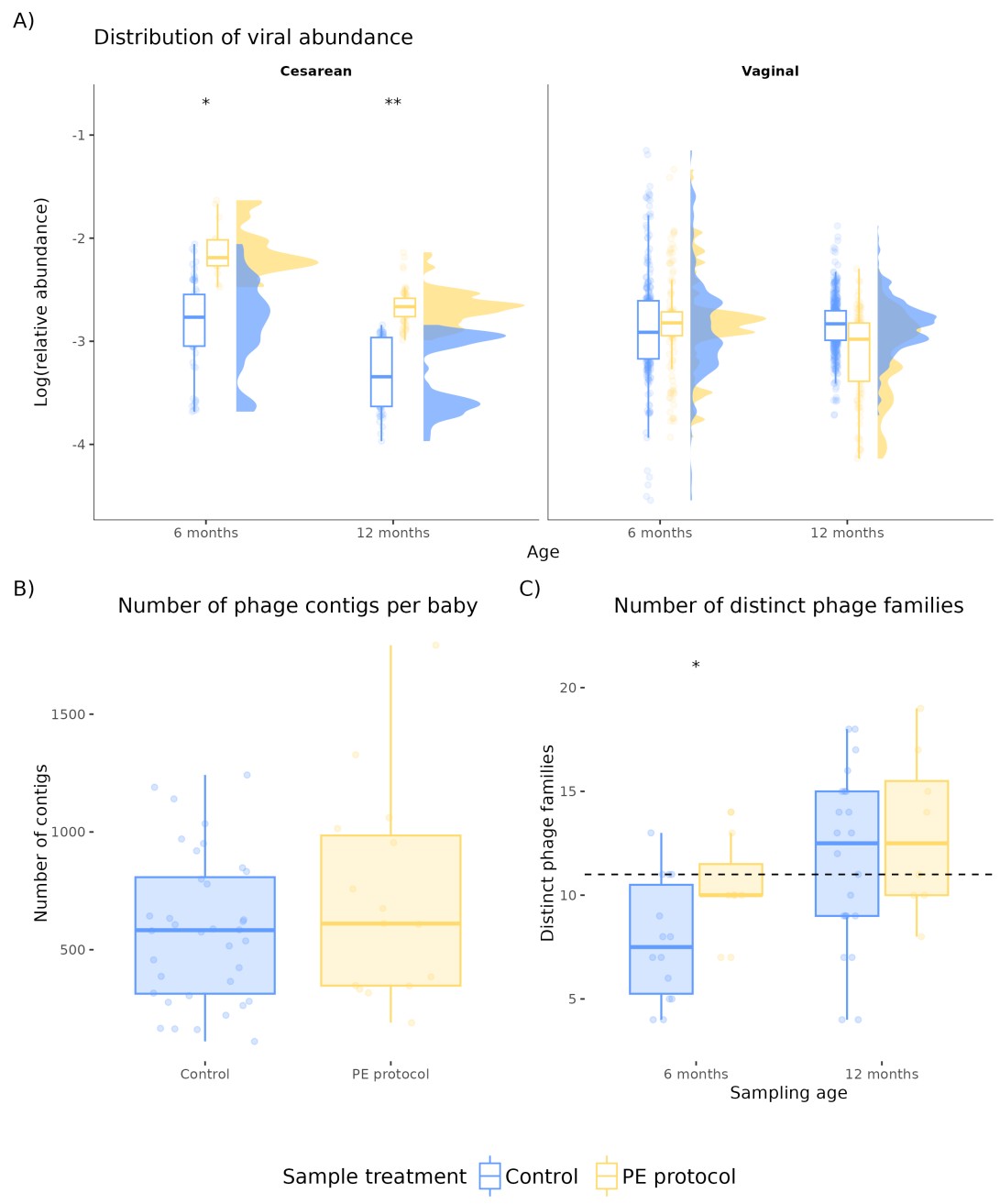

**FIG 2** Phage abundance overview. Samples of 1 month of age have no control counterpart. Thus, no comparison was performed at that sample age. (A) Viral abundance distribution over all infant samples in terms of their logarithm of the relative abundance by delivery and age. (B) Number of viral contigs found per infant. (C) Number of unique viral families detected. Comparisons were performed using LMM. ns: not significant, *: $P < 0.05$, **: $P < 0.01$

associations between viral taxa and host-related factors (Fig. 5; Table S3). In PE-treated samples, age-associated prevalence patterns emerged more frequently across multiple phage families ($P < 0.05$, Table S3). In contrast, these associations were less commonly observed in control samples.

### Phage–bacterial association in the infant gut

Bacterial hosts were predicted using the iPHoP framework (36) and Kraken2 (32) with the HGDB (23). In total, 25,862 contigs (78.23%) were assigned to bacterial hosts at the genus level, and 8,554 contigs (25.87%) at the species level. To explore host-specific

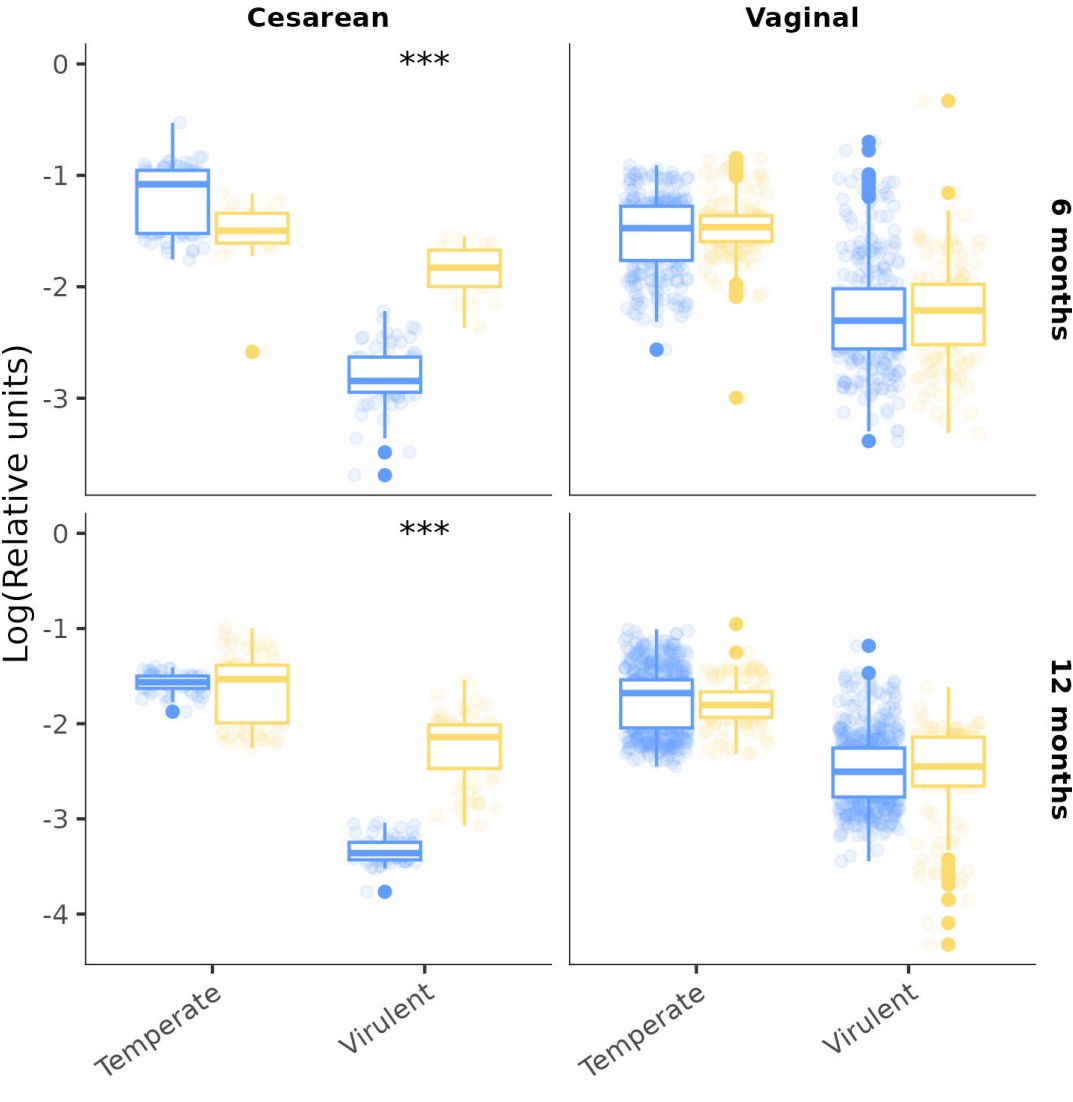

**FIG 3** Comparison of log-transformed viral relative units of different phage life cycle types (temperate and virulent) using LMM under two conditions: Control (blue) and PE protocol (yellow). Two delivery methods are shown (cesarean and vaginal) and at two time points (6 and 12 months). No taxonomy was considered. ***: $P < 0.001$.

phage dynamics, we applied LMM to phage abundances stratified by bacterial genus. PE protocol revealed numerous associations between phage-host pairs and infant variables, such as age and delivery mode, that were not detectable in control samples (Fig. 6; Table S3).

## DISCUSSION

In this study, we optimized and validated a PEG-based protocol for phage DNA enrichment from infant fecal samples and assessed its effect on the observed gut phageome of 41 infants at 6 and 12 months of age. To maximize resource efficiency, we

pooled viral and bacterial DNA for co-sequencing, thereby creating a cost-effective and scalable pipeline for comprehensive metagenomic profiling. The protocol was developed with the explicit aim of enhancing the detectability of viral signatures, especially in low-volume samples where traditional approaches underperform.

We piloted five protocol variations grounded in two commonly used enrichment strategies—DTT-based and PEG-based precipitation methods (7, 17). Our focus was not

### A    Distribution of alpha diversity

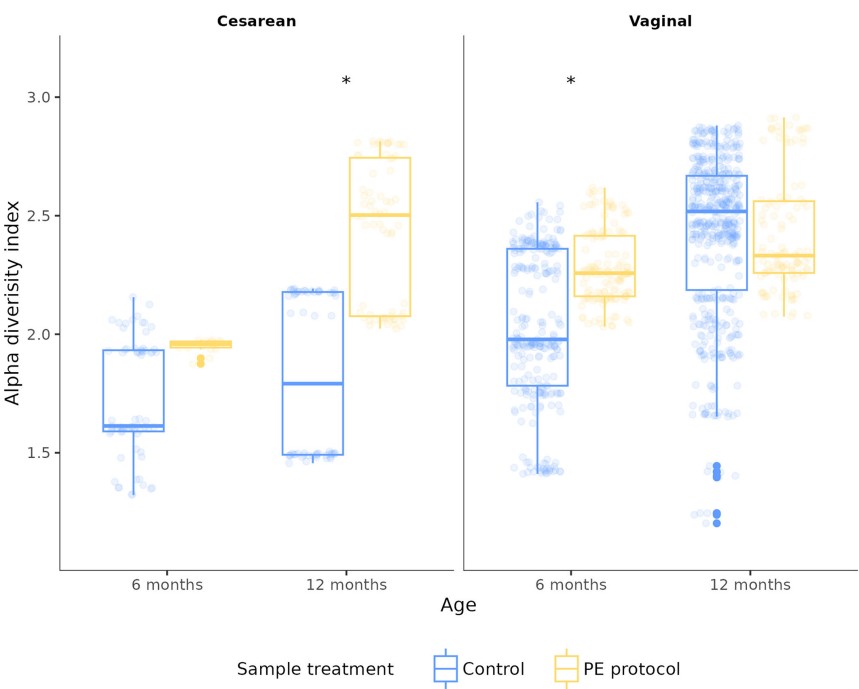

### B

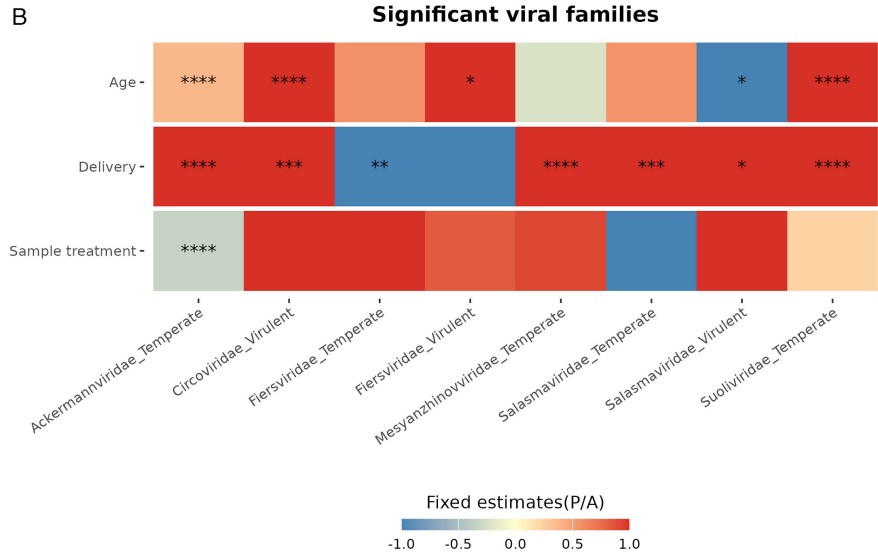

**FIG 4**   (A) Distribution of phage's alpha diversity index. LMMs were applied under two conditions: Control (blue) and PE protocol (yellow). Two delivery methods are shown (cesarean and vaginal) and at two time points (6 and 12 months). (B) Significant associations between viral families and their influential factors: age, delivery method, and sample treatment. LMM with binomial distribution was applied to three factors displayed on the rows (age, delivery, and sample treatment), while the columns represent specific viral families. Fixed estimates were cutoff between −1 and 1 for better visualization. *: $P < 0.05$, **: $P < 0.01$, ***: $P < 0.001$, ****: $P < 0.0001$.

on direct methodological comparison but on optimizing a PEG-based workflow that performs reliably across the constraints of infant sampling. The refined PEG protocol (P3) yielded the highest DNA concentrations with consistent performance, particularly in older infants (12 months), underscoring the necessity of age-appropriate optimization in light of evolving fecal matrix composition and microbiota maturation.

Our findings confirmed that temperate phages—commonly integrated into bacterial genomes—constituted the majority of phage sequences, aligning with prior studies (42, 43). However, virulent phages were also present in significant proportions and were more likely to be overlooked without enrichment. This underestimation was especially apparent in cesarean-born infants, potentially biasing comparisons between clinical subgroups if not corrected by enrichment. These discrepancies may stem from limitations in current bioinformatic classifiers, the dynamic nature of virulent phage activity, or host-specific influences on detectability (13, 14, 16).

Applying the PE treatment influenced the observed prevalence, diversity, and composition of the infant gut phageome, especially among the cesarian born infants. Without the PE treatment, there was a clear underestimation of the relative abundance of virulent phages among the cesarian born infants, but not among the vaginally born. A larger fraction of the phageome was present as free viral particles in cesarean-born infants compared with vaginally born infants, making the effect of PE treatment more pronounced in the former. This underscores the importance of applying PE when comparing groups, since omitting it can underestimate the phageome and introduce systematic bias. Such bias may obscure biological signals and add noise to the data. In line with this, PE-treated samples revealed stronger associations of the phageome with both age and birth mode than untreated samples.

At the host level, the PE treatment revealed that vaginally born infants had a higher prevalence of phages targeting Bacteroides, Bifidobacterium, Phocaeicola, and Veillonella genera, while cesarian born infants had a higher prevalence of phages targeting bacteria genera, such as *Streptococcus*, *Clostridium*, and *Enterocloster*, which is likely due to the birth-mode associated differences in the bacterial composition. These results are in line with previous knowledge that delivery mode exerts a lasting influence on the development of phage-bacterial relationships, due to differences in early microbial exposure (4). Further analyses on key genes could extend the resolution of these findings and microbe–phage links.

In conclusion, our findings demonstrate that the decision to use PE is not merely a technical detail but a critical determinant of biological interpretation. The omission

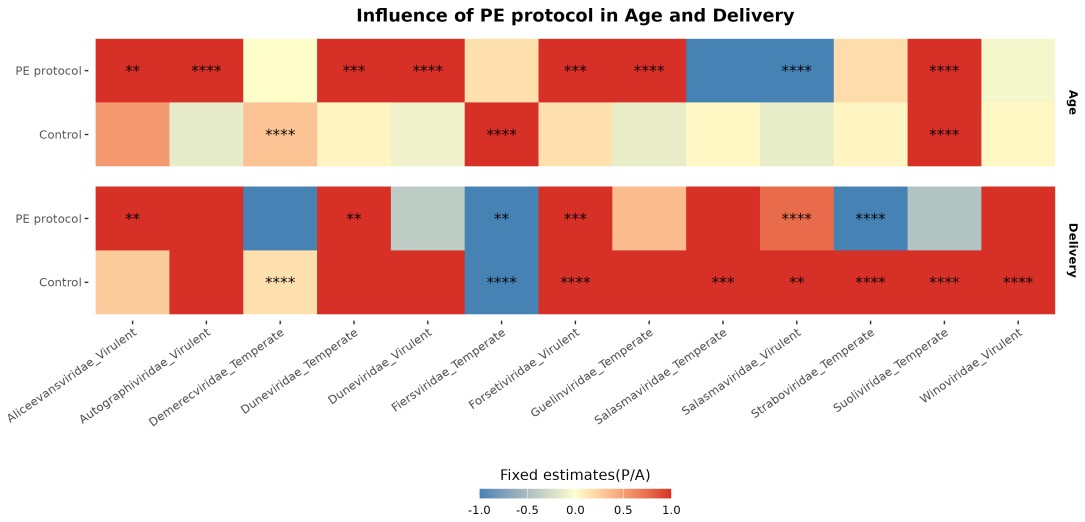

**FIG 5** Heatmap of LMM fixed estimates using binomial distribution. The heatmap shows the influence of PE protocol in age and delivery mode over different phage families; missing and non-resolved taxa, in contrast, were removed in order to compare pure viral families stratified by life cycle. Fixed estimates were cutoff between −1 and 1 for better visualization. **: *P* < 0.01, ***: *P* < 0.001, ****: *P* < 0.0001.

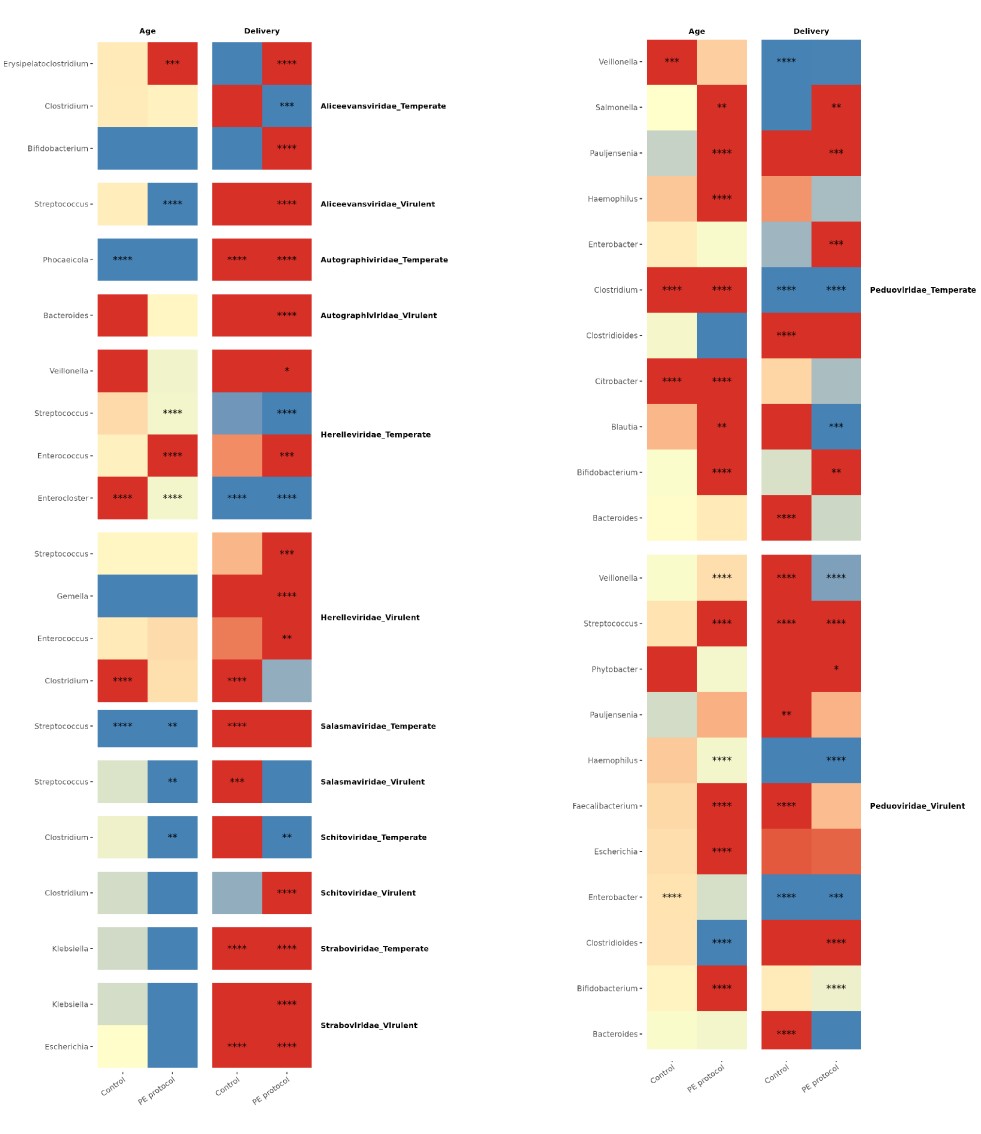

## Significant phage's Host

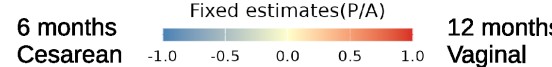

**Fixed estimates(P/A)**

6 months — 12 months
Cesarean — -1.0  -0.5  0.0  0.5  1.0 — Vaginal

**FIG 6** Associated hosts of phage families using LMM. A heatmap indicates the fixed estimates and the statistical significance for age and delivery, with blue representing 6 months/cesarean and red representing 12 months/vaginal, respectively. Fixed estimates were cutoff between −1 and 1 for better visualization. *: $P < 0.05$, **: $P < 0.01$, ***: $P < 0.001$, ****: $P < 0.0001$.

of PE can lead to substantial underrepresentation of key viral taxa linked to age and delivery mode. In infant studies, where fecal composition and microbial dynamics are highly variable and sensitive to early-life exposures, without enrichment, the group comparisons may yield misleading conclusions due to differential recovery of phage populations across subgroups. Conversely, the use of an optimized PE protocol improves the resolution and sensitivity of virome profiling, providing a more consistent and representative view of virus–bacteria–host interactions. As the field moves toward higher-resolution studies of the gut virome, especially in early life, methodologically

rigorous approaches will be essential for elucidating the complex interplay between phages, bacteria, and host development.

## ACKNOWLEDGMENTS

This work was supported by ERC Starting Grant 101039583 - MICROECO.

## AUTHOR AFFILIATIONS

[1]Human Microbiome Research Program, Faculty of Medicine, University of Helsinki, Helsinki, Finland

[2]Department of Medical Microbiology, Infectious Diseases and Infection Prevention, School of Nutrition and Translational Research in Metabolism (NUTRIM), Maastricht University Medical Centre+, Maastricht, the Netherlands

[3]Department of Pediatrics, University of Helsinki and Children's Hospital HUS, Helsinki, Finland

[4]Department of Bacteriology and Immunology, Faculty of Medicine, University of Helsinki, Helsinki, Finland

## AUTHOR ORCIDs

Sandro Valenzuela-Diaz  http://orcid.org/0000-0002-2284-8243
Saija Kiljunen  http://orcid.org/0000-0003-0461-7270
Anne Salonen  http://orcid.org/0000-0002-6960-7447
Katri Korpela  http://orcid.org/0000-0001-5031-2713

## AUTHOR CONTRIBUTIONS

Katri Korpela, Conceptualization, Investigation, Methodology, Project administration, Resources, Supervision, Validation, Visualization, Writing – review and editing.

## DATA AVAILABILITY

Read files with viral data gathered from infant samples are deposited under the ENA project accession PRJEB89462. Scripts and tables: https://github.com/Sanrrone/Impact-of-phage-enrichment-on-the-observed-infant-gut-phageome. Contigs: https://github.com/Sanrrone/Impact-of-phage-enrich-ment-on-the-observed-infant-gut-phageome/tree/main/contigs. Samples: Viral mapped reads are publicly available under ENA (ID PRJEB89462). Scripts, assemblies, and supplementary figures and tables are backed up under a unique DOI: https://doi.org/10.5281/zenodo.15481575.

## ETHICS APPROVAL

The research was approved by the Helsinki University Hospital Ethics Committee (HUS/2346/2016). Informed consent was obtained from the parents of the study subjects.

## ADDITIONAL FILES

The following material is available online.

### Supplemental Material

**Supplemental figures (Spectrum02153-25-s0001.pdf).** Figures S1 to S4.
**Table S1 (Spectrum02153-25-s0002.csv).** Summary of the experiments performed to scale DTT and PEG protocols.
**Table S2 (Spectrum02153-25-s0003.csv).** Metadata of the samples used in the study.
**Table S3 (Spectrum02153-25-s0004.xlsx).** Statistics performed.

Open Peer Review

**PEER REVIEW HISTORY (review-history.pdf).** An accounting of the reviewer comments and feedback.

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
