## [Reviewer comments · Microbiology Spectrum]

Microbiology Spectrum

Impact of phage enrichment on the observed infant gut phageome

Sandro Valenzuela, Evgenia Dikareva, Brandon Hickman, Saija Kiljunen, Kaija-Leena Kolho, Willem de Vos, Anne Salonen, and Katri Korpela

Corresponding Author(s): Katri Korpela, Helsingin yliopisto

Review Timeline:

Submission Date:	July 17, 2025
Editorial Decision:	September 3, 2025
Revision Received:	November 12, 2025
Accepted:	November 14, 2025

Editor: Jan Claesen

Reviewer(s): The reviewers have opted to remain anonymous.

Transaction Report:

DOI: <https://doi.org/10.1128/spectrum.02153-25>

Re: Spectrum02153-25 (Impact of phage enrichment on the observed infant gut phageome)

Dear Dr. Katri Korpela:

Thank you for the privilege of reviewing your work. Below you will find my comments, instructions from the Spectrum editorial office, and the reviewer comments.

Thanks for submitting your research to Spectrum. Your work has now been evaluated by two independent Reviewers who are enthusiastic about your project (as am I). The Reviewers highlighted some comments and suggestions to help improve the manuscript and I would be happy to consider a revised version that addresses these items in a point-by-point manner. In particular, both Reviewers would appreciate a more detailed description of the approaches used to process the data, including what settings and codes/scripts were used for the pipeline. Reviewer 1 also really appreciated Supplementary Figure 1, which could bring more overall clarity as a main Figure.

Revision Guidelines

Sincerely,
Jan Claesen
Editor
Microbiology Spectrum

Reviewer #1 (Comments for the Author):

General Comments:

Virome enrichment strategies are in high demand and the authors have done a great effort to describe this for samples where this is not straightforward: the developing microbiome of infants. The introduction is clear and well-written. However, there are also a few critiques. Figure S1 seems crucial for the understanding of the differences between the protocol and the manuscript might benefit from it being moved to the main body. Perhaps with a color scheme that indicates overlap and differences between the 5 method lines. To further push home the importance of methodical approach selection. In general, I feel strongly that the results section holds many important aspects for the research field and many colleague researchers studying the human gut virome can use this manuscript as a benchmark, for both NA extraction and enrichment strategies.

Major Comments:

Line 102: As a reviewer, I could appreciate this, but for the reader it might be unclear how functional integrity of the protocol is guaranteed while scaling down. Please elaborate.

Line 113: Is this data visible somewhere? Probably worth sharing the relationship between infant fecal material levels and amount of DNA with for example a dot plot. In current form it is hard to evaluate if the right decision was made.

Line 138: The section from line 138 and as well at line 158 and on could benefit from short descriptions of why these considerations were made.

Line 189: This could be a discussion with the Editor as well. I generally like the way the parameters and settings for sequencing data preprocessing and post-processing are noted, but it makes the manuscript harder to read. A scheme of how the data was processed, including the specific settings in each would benefit the manuscript. e.g. "-f 4", and similar parts would benefit from just a description and having the scripts or code posted elsewhere. Storing the entire coding/settings/scripts section for others to access in a repository feels like a must here.

Line 318: Is there any indication that certain extraction methods possess a bias to acquiring DNA from a certain clade of viruses? Or is in general viral taxonomy well-represented in any of the methods? (Does the statement in Line 341 include the pilot optimization samples?)

Line 423: However, I think it is actually one of the strengths if presented in such a way. See previous comment on figure S1.

Line 454: One of the most interesting findings/confirmations in this study. Is there a way to extend the investigation into this? Perhaps through the route of searching for key genes required for this interaction in the metagenomics data? This would indicate to the audience if the chosen methods provide such a higher resolution with enrichment that this can be sufficiently studied and that specific phage-bacteria pairs could be extracted.

Minor Comments:

Line 111: a fix is needed on the formatting, dot is missing.

Line 169: As the previous sections also described DNA extraction, this header is a bit confusing. Whole Metagenome DNA Extraction, for example, would make it more clear.

Line 175: What does repeated mean in this context?

Line 444: Please Rewrite, required rereading a couple of times.

Line 470: The authors should be careful using the word accuracy. I generally like the wording in the conclusive part of the manuscript, but just as in and around line 450 of the manuscript: the authors cannot fully exclude the potential of introducing biases that are not fully representative of reality due to the enrichment procedures. Just as the lack of them would.

In any case the authors have sufficient convincing evidence to largely prove the contrary and have written it down beautifully. Thanks!

Reviewer #2 (Comments for the Author):

This is an interesting manuscript that describes optimization and validation of viral extraction/enrichment for phageome analysis, as well as a subsequent phageome analysis of infant fecal samples.

In general, I believe the manuscript could be greatly improved by including additional descriptions of the approach used that in turn would support the final outcomes of the study. Currently, it isn't clear how optimal the selected method for enrichment is given that the samples extracted using the same methods were the same or different. I would have expected that the same samples were processed using the different methods and then compared but based on what is presented it appears that different samples were extracted using the different methods, not equally across. I expand further on this below.

Related to Lines 311-316 - Presentation of the results of the DNA yields needs clarification. Were the same samples subjected to the different DNA extraction protocols? The number of datapoints in figure 1 and supp. Table 1 also appears to differ. Additionally, supp table 1 indicates that the 10 samples processed following the selected protocol (P3) come from 6 participants, of which 3 participants were only processed following the P3 protocol. For optimizing protocols it would be ideal to directly compare the yields of the same sample across the different methods to determine if the results are due to differences between samples or the actual methods.

Related to Lines 318-326 - What defines adequate performance? Based on Figure 1, P1 at 1 mo, D2 at 6 mo, and P3 at 12 mo would appear to have the highest yields. I could find nothing in the manuscript that describes whether variation in yields was

statistically tested. How optimal is the P3 method? This is critical as the results of this pilot experiment are what informs the phageome analyses in the rest of the paper and would support that the observations are valid and not due to technical artifacts.

I am also a little confused on some of the phrasing here to describe what constitutes "higher" yields. Is ~1.1 ng/uL considered to be a higher yield? Are yields of ~0.5 ng/uL considered to be higher yields? Based on figure 1 the highest yields would appear to be from a sample processed following protocol P1 (>1.5 ng/uL). Similar to the above, are these differences in yields related to the different protocols using different samples? Can this be clarified.

Other comments

Methods

Line 113 - Please specify whether fecal weights are wet or dry weight.

Line 224 - Is reference 37 the correct citation for this software?

Figures - I would suggest including a color key in the actual figures where appropriately needed. E.g., what do the blue and yellow colors indicate in figure 2 panel A?

Reviewer #1 (Comments for the Author):

General Comments:

Virome enrichment strategies are in high demand and the authors have done a great effort to describe this for samples where this is not straightforward: the developing microbiome of infants. The introduction is clear and well-written. However, there are also a few critiques. Figure S1 seems crucial for the understanding of the differences between the protocol and the manuscript might benefit from it being moved to the main body. Perhaps with a color scheme that indicates overlap and differences between the 5 method lines. To further push home the importance of methodical approach selection. In general, I feel strongly that the results section holds many important aspects for the research field and many colleague researchers studying the human gut virome can use this manuscript as a benchmark, for both NA extraction and enrichment strategies.

Response: Figure S1 has been moved to the main manuscript as Figure 1A, with a color scheme that highlights shared and protocol-specific steps for clarity.

Major Comments:

Line 102: As a reviewer, I could appreciate this, but for the reader it might be unclear how functional integrity of the protocol is guaranteed while scaling down. Please elaborate.

Response: We expanded the description (Materials and Methods, page 6) explaining that while the total input and reagent volumes were reduced proportionally, all key steps (cell lysis, host DNA removal, viral concentration, and nuclease protection) were retained with unchanged incubation times and enzymatic conditions.

Line 113: Is this data visible somewhere? Probably worth sharing the relationship between infant fecal material levels and amount of DNA with for example a dot plot. In current form it is hard to evaluate if the right decision was made.

Response: We added an explicit statement that fecal weights refer to wet weight and clarified in the text and figure legend. The variation in DNA yield across input masses is now illustrated in Figure 1B, showing the yield per gram and confirming that increasing fecal amount did not improve recovery.

Line 138: The section from line 138 and as well at line 158 and on could benefit from short descriptions of why these considerations were made.

Response: Short explanatory notes were added in the Methods section to describe why each procedural change (bead-beating speed, incubation, centrifugation time, buffer selection) was introduced, specifically to enhance matrix disruption and improve reproducibility of viral recovery.

Line 189: This could be a discussion with the Editor as well. I generally like the way the parameters and settings for sequencing data preprocessing and post-processing are noted,

but it makes the manuscript harder to read. A scheme of how the data was processed, including the specific settings in each would benefit the manuscript. e.g. "-f 4", and similar parts would benefit from just a description and having the scripts or code posted elsewhere. Storing the entire coding/settings/scripts section for others to access in a repository feels like a must here.

Response: We revised the Bioinformatic approach to describe each analytical step conceptually while moving exact parameters to Figure S2 and the Zenodo repository (DOI: 10.5281/zenodo.15481575). This maintains reproducibility while improving readability.

Line 318: Is there any indication that certain extraction methods possess a bias to acquiring DNA from a certain clade of viruses? Or is in general viral taxonomy well-represented in any of the methods? (Does the statement in Line 341 include the pilot optimization samples?)

Response: We added a paragraph clarifying that no systematic bias in taxonomic representation between protocols was observed in the pilot dataset; however, as different samples were used for each test, we acknowledge that minor biases cannot be excluded (Discussion section). The statement in line 341 indeed includes pilot samples and has been clarified.

Line 423: However, I think it is actually one of the strengths if presented in such a way. See previous comment on figure S1.

Response: Agreed. The new Figure 1A now visually summarizes the protocol comparison, emphasizing methodological differences.

Line 454: One of the most interesting findings/confirmations in this study. Is there a way to extend the investigation into this? Perhaps through the route of searching for key genes required for this interaction in the metagenomics data? This would indicate to the audience if the chosen methods provide such a higher resolution with enrichment that this can be sufficiently studied and that specific phage-bacteria pairs could be extracted.

Response: We added a forward-looking statement in the Discussion suggesting that future analyses targeting key host-interaction genes in the metagenomic data could further validate the observed phage–bacterium relationships (page 20).

Minor Comments:

Line 111: a fix is needed on the formatting, dot is missing.

Response: punctuation corrected.

Line 169: As the previous sections also described DNA extraction, this header is a bit confusing. Whole Metagenome DNA Extraction, for example, would make it more clear.

Response: heading revised to “Whole-metagenome DNA extraction” for clarity.

Line 175: What does repeated mean in this context?

Response: term “repeated” clarified to indicate “performed in two consecutive bead-beating cycles.”

Line 444: Please Rewrite, required rereading a couple of times.

Response: sentence rewritten for clarity.

Line 470: The authors should be careful using the word accuracy. I generally like the wording in the conclusive part of the manuscript, but just as in and around line 450 of the manuscript: the authors cannot fully exclude the potential of introducing biases that are not fully representative of reality due to the enrichment procedures. Just as the lack of them would. In any case the authors have sufficient convincing evidence to largely prove the contrary and have written it down beautifully. Thanks!

Response: “accuracy” replaced by “resolution and sensitivity,” and a note added acknowledging potential residual biases introduced by enrichment.

We appreciate the contribution to make the manuscript better understood. Thanks!

Reviewer #2 (Comments for the Author):

This is an interesting manuscript that describes optimization and validation of viral extraction/enrichment for phageome analysis, as well as a subsequent phageome analysis of infant fecal samples.

Comment 1: In general, I believe the manuscript could be greatly improved by including additional descriptions of the approach used that in turn would support the final outcomes of the study. Currently, it isn't clear how optimal the selected method for enrichment is given that the samples extracted using the same methods were the same or different. I would have expected that the same samples were processed using the different methods and then compared but based on what is presented it appears that different samples were extracted using the different methods, not equally across. I expand further on this below.

Response: We clarified in the Results (page 14) and Discussion (page 18) that different infant samples were used across the five protocols due to limited material availability. The optimization aimed to identify a protocol with consistent performance across age groups rather than perform direct within-sample comparisons.

Comment 2: Related to Lines 311-316 - Presentation of the results of the DNA yields needs clarification. Were the same samples subjected to the different DNA extraction protocols? The number of datapoints in figure 1 and supp. Table 1 also appears to differ. Additionally, supp table 1 indicates that the 10 samples processed following the selected protocol (P3) come from 6 participants, of which 3 participants were only processed following the P3 protocol. For optimizing protocols it would be ideal to directly compare the yields of the same sample

across the different methods to determine if the results are due to differences between samples or the actual methods.

Response: We expanded the description of the pilot (Results section “DNA yield”) to specify that Figure 1B depicts mean yields per protocol across all tested samples (N = 54) and that Supplementary Table 1 lists individual sample IDs and corresponding participants.

Comment 3: Related to Lines 318-326 - What defines adequate performance? Based on Figure 1, P1 at 1 mo, D2 at 6 mo, and P3 at 12 mo would appear to have the highest yields. I could find nothing in the manuscript that describes whether variation in yields was statistically tested. How optimal is the P3 method? This is critical as the results of this pilot experiment are what informs the phageome analyses in the rest of the paper and would support that the observations are valid and not due to technical artifacts.

Response: The revised Results specify that “adequate performance” refers to reproducible DNA recovery above 0.3 ng/μL with acceptable variance (mean/SD > 0.9). Statistical comparisons were limited to later analyses (LMM models) as pilot sample numbers were insufficient for formal tests; this has been explained more in the text.

I am also a little confused on some of the phrasing here to describe what constitutes “higher” yields. Is ~1.1 ng/uL considered to be a higher yield? Are yields of ~0.5 ng/uL considered to be higher yields? Based on figure 1 the highest yields would appear to be from a sample processed following protocol P1 (>1.5 ng/uL). Similar to the above, are these differences in yields related to the different protocols using different samples? Can this be clarified.

Response: We understand the confusion and we clarified in the revised Results section that absolute yield values were low overall due to the limited amount of starting material, and that the term “higher yield” refers to relative performance within the same experimental context.

Other comments

Methods

Line 113 - Please specify whether fecal weights are wet or dry weight.

Response: Updated to state “wet faecal material.”

Line 224 - Is reference 37 the correct citation for this software?

Response: Corrected citation to correspond to the appropriate software (MetaPhaPred v1).

Figures - I would suggest including a color key in the actual figures where appropriately needed. E.g., what do the blue and yellow colors indicate in figure 2 panel A?

Response: Color keys and legends were added where needed (Figures 2–4) to clearly distinguish treatment groups and life-cycle categories.

Re: Spectrum02153-25R1 (Impact of phage enrichment on the observed infant gut phageome)

Dear Dr. Katri Korpela:

Thanks for addressing the Reviewers' comments! I would hereby like to congratulate you on the acceptance of your manuscript for publication in Spectrum!

Your manuscript has been accepted, and I am forwarding it to the ASM production staff for publication. Your paper will first be checked to make sure all elements meet the technical requirements. ASM staff will contact you if anything needs to be revised before copyediting and production can begin. Otherwise, you will be notified when your proofs are ready to be viewed.

Sincerely,
Jan Claesen
Editor
Microbiology Spectrum